

# Functional ecological traits in young and adult thalli of canopy-forming brown macroalga *Gongolaria barbata* (Phaeophyta) from a transitional water system

Maria Luisa Pica[1], Ermenegilda Vitale[1,2], Rosa Donadio[1], Giulia Costanzo[1], Marco Munari[3,4], Erika Fabbrizzi[1,2,5], Simonetta Fraschetti[1,2,5] and Carmen Arena[1,2]

[1] Department of Biology, University of Naples Federico II, Naples, Italy
[2] NBFC, National Biodiversity Future Center, Palermo, Italy
[3] Department of Integrative Marine Ecology, Stazione Zoologica Anton Dohrn, Napoli, Italy
[4] Department of Biology, Stazione Idrobiologica 'Umberto d'Ancona', University of Padova, Padova, Italy
[5] Consorzio Nazionale Interuniversitario per le Scienze del Mare, Roma, Italy

Corresponding author
Carmen Arena, c.arena@unina.it

## ABSTRACT

**Background**. *Gongolaria barbata* is a canopy-forming brown macroalga that thrives in the intertidal and subtidal habitats of the warm-temperate Mediterranean Sea, which is particularly exposed to environmental changes due to its peculiar geographical location and exposure to both global and local stressors. Testing whether this species is featured by specific functional, eco-physiological and biochemical traits allowing an efficient use of habitat resources and adaptation to environmental stress, and whether this potential might change with population growth, is essential for predicting the performance of the algae under different environmental abiotic variables (*e.g.*, temperature, nutrient availability, light) and biotic interactions (such as grazing).

**Methods**. Young (juveniles) and adult thalli of *G. barbata* were sampled in the winter season from the Venice Lagoon, Italy, featured by high environmental changes (temperature, salinity) and analyzed for thallus dry matter content (TDMC), photosynthetic activity, photosynthetic pigment and protein content, and antioxidant capacity to assess if thallus age may be considered a significant driver in determining the ecological responses of this species to environmental changes.

**Results**. Our results showed that TDMC was higher in adults than juveniles. At the functional level, rapid light curves indicated an elevated photosynthetic efficiency in juveniles compared to adults highlighted by the higher quantum yield of PSII electron transport, electron transport rate, and Rubisco content observed in juveniles. On the contrary, adults exhibited a higher non-photochemical quenching and total pigment concentration. No difference in maximum PSII photochemical efficiency and D1 protein content between the two thalli groups was found. Along with better photosynthesis, juveniles also displayed a higher amount of total polyphenols, flavonoids, and tannins, and a stronger antioxidant capacity compared to adults.

**Conclusions**. Our findings revealed significant differences in the eco-physiological characteristics of *G. barbata* at different growth stages. It was observed that young thalli,

allocate more energy to photosynthesis and chemical defenses by increasing the production of antioxidant compounds, such as polyphenols, flavonoids, and tannins. With growth, thalli likely adopt a more conservative strategy, reducing photosynthesis and promoting structural biomass accumulation to mitigate the potential risks associated with prolonged exposure to environmental stressors, such as the wavy way. Although our study focused on a single phase of *G. barbata* life cycle under winter settings, it offers preliminary insights into this species eco-physiological traits and auto-ecology. Future research could explore the potential implications of these findings, evaluating the species' resilience to environmental changes at the population level.

## INTRODUCTION

Canopy-forming brown macroalgae (*i.e.,* kelps, fucoids), are well-known for their crucial role as habitat formers in both intertidal and subtidal habitats of cold-temperate latitudes. They increase the three-dimensional complexity and spatial heterogeneity of the substrate they colonize (*Verdura et al., 2018*). Their vertical and branched canopies increase coastal primary production, offer shelter to smaller epiphytic algae and many meiofaunal invertebrates, represent nursery areas for juvenile fish, and protect them from predators and hydrodynamics (*Krumhansl et al., 2016*; *Verdura et al., 2018*; *Orlando-Bonaca, Pitacco & Lipej, 2021*; *Gran et al., 2022*; *Manca et al., 2022*). The conservation status of these long-living species is indicative of habitat loss, environmental degradation (*Orlando-Bonaca, Pitacco & Lipej, 2021*) and quality of Mediterranean coastal waters (*Ballesteros et al., 2007*; *Orlando-Bonaca et al., 2013*). Finally, they also contribute to many ecosystem services by providing foraging and preserving species of commercial interest, sustaining coastal fisheries, absorbing pollutants and filtering water, reoxygenating sediments and acting as an important sink for carbon through its sequestration to the seafloor; hence, they are known to be one of the most productive ecosystems on Earth (*Gran et al., 2022*; *Manca et al., 2022*). Regardless of all benefits, these communities are exposed to multiple stressors and threatened by human activities, including eutrophication, pollution, outbreaks of grazers caused by overfishing, invasive species introduction, increasing sediment resuspension and climate change-driven consequences (*Ballesteros et al., 2007*; *Orlando-Bonaca et al., 2013*; *Orlando-Bonaca, Pitacco & Lipej, 2021*). Species belonging to the genus *Cystoseira sensu lato* (*s.l.*) (*Molinari-Novoa & Guiry, 2020*) are endemic of the Mediterranean, classified as threatened (except for *C. compressa*) under the Barcelona Convention (Annex II of the Barcelona Convention, COM/2009/0585/FIN), and protected by local regulations. Despite the reduction of impacts imposed by legislation, *Cystoseira s.l.* forests experienced regression events at the basin scale that led to habitat loss (*Cebrian et al., 2021*; *Verdura et al., 2023*), and, in some cases, to regime shift to algal turfs, which are less complex

and poorly productive communities inhibiting recolonization by canopy-forming species (*Benedetti-Cecchi et al., 2015*).

In recent years, several studies have been conducted to assess the ecological strategies of marine plants and macroalgae in terms of resource-use strategies by evaluating physiological and structural attributes (*Starko & Martone, 2016*; *Ishizawa et al., 2021*; *Sakanishi, Kasai & Tanaka, 2023*). Understanding whether a species exhibits variations in functional traits, such as eco-physiological and biochemical characteristics across different growth stages is pivotal for evaluating its potential ecological adaptation or vulnerability to future changes in dynamic environmental conditions.

Macroalgae have developed specific eco-physiological, biochemical, morphological mechanisms for mitigating fluctuations in environmental factors, demonstrating spatial and temporal adaptations and species-specific responses to single or combined stressors (*Hurd et al., 2014*; *Starko & Martone, 2016*; *Cappelatti, Mauffrey & Griffin, 2019*; *Ishizawa et al., 2021*; *Hanley, Firth & Foggo, 2024*). The thallus age, as plant ontogeny in terrestrial ecosystems (*Rusman et al., 2020*), could represent a valuable feature in assessing growth-defense mechanisms against environmental stressors (*Pellizzari, Oliveira & Yokoya, 2008*), including unpredictable events, such as temperature rise, tidal variations, nutrient and salinity fluctuations, water acidification and grazing pressure. However, to date, the adaptation patterns of canopy-forming macroalgae across different ages are rarely investigated, limiting our potential to predict different vulnerability across life stages with consequences at population levels.

This study focused on juvenile and adult individuals of *Gongolaria barbata* (Stackhouse) Kuntze, a Mediterranean widespread canopy-forming macroalga. *Gongolaria barbata* is a well-known adapted species to both euryhaline and polyhaline environments (*Sadogurska et al., 2021*; *Tursi et al., 2023*), and even broad changes in salinity do not affect its growth (*Baghdadli et al., 1990*; *Irving et al., 2009*). Studies on *Cystoseira s.l* demonstrated good tolerance of this species to acidification, which promoted growth rate, photosynthesis, antioxidant activity, and photoprotection (*Celis-Plá et al., 2017*). *Gongolaria barbata* also showed adaptability to a wide range of temperatures (*Orfanidis, 1991*; *Iveša et al., 2022*; *Fabbrizzi et al., 2023*). However, while adults have been extensively studied, there is missing information about the ecophysiology of early life stages for this species. Adults in the vegetative phase can endure high temperatures up to 30–34 °C during summer and freezing temperatures, during winter (*Iveša et al., 2022*). Recruits develop optimally at 15 °C and sufficiently from 10 °C to 25 °C (*Orfanidis, 1991*) but are particularly sensitive to temperature-induced stress, experiencing high mortality rates (*Lokovšek et al., 2024*).

Grazing also may threaten the large-scale population and restoration interventions of *G. barbata* (*Savonitto et al., 2021*). Herbivores and thallus age deeply influence grazers' feeding preferences, which depend on the different palatability resulting from chemical and morphological changes occurring during thalli development (*Van Alstyne, Ehlig & Whitman, 1999*).

The concentration of some compounds, such as polyphenols, varies not only in response to seasonal shifts, nutrient levels, acidification, temperature fluctuations, irradiance intensity or desiccation (*Celis-Plá et al., 2014*; *Celis-Plá et al., 2016*; *Celis-Plá et al., 2017*)

but also with thallus growth, morphology and age. Indeed, adult thalli concentrate in longer, more complex, and degradable forms than younger individuals. At the same time, a higher amount of polyphenols and phlorotannins is generally produced for the protection of the zygotes or in the cell-wall hardening to cope with grazers (*Mannino & Micheli, 2020*; *Monserrat et al., 2023*). As age increases, tissues tend to thicken, providing more resistance to grazing and physical stresses (*Mauffrey, Cappelatti & Griffin, 2020*).

Our study explores essential eco-physiological, biochemical and functional traits, such as photosynthetic activity, antioxidant defenses, and thallus dry matter content, to investigate potential age-related variations in juvenile *vs* adult individuals of *Gongolaria barbata* from the transitional water system of the Venice Lagoon (Italy). By examining how these attributes may change with age and influence local adaptation mechanisms, this study may shed light on the trade-off between acquisition and conservation strategies employed by young and adult individuals of *G. barbata* populations. Our findings may offer valuable insights into the physiological cost associated with thalli growth and development, defense mechanisms, and overall primary production (*Cappelatti, Mauffrey & Griffin, 2019*; *Sakanishi, Kasai & Tanaka, 2023*).

## MATERIALS & METHODS

### Sampling and experimental design

Juveniles and adult individuals ($n = 5$ per group) of the species *G. barbata* were randomly sampled in winter, at the beginning of February 2023, before the reproductive phase (*Bevilacqua et al., 2019*; *Savonitto et al., 2021*). Sampling was performed in Italy (Fig. 1A), at the offshore location of Ca' Roman (45°14′42.2″N 2°17′44.7″E) (Fig. 1B), recognized as natural reserve of regional interest under the Regional Law n. 40/1984 and situated within the Venice Lagoon (Natura 2000 site, IT 3250023) (Fig. 1). The permission for sampling was provided by Regione del Veneto (decree number 369, date 04.05.2023).

Juveniles and adult thalli were collected at a depth of two meters, in a sampling area of 10 m$^2$, at 0.20–1.0 m from each other and at a sea-water temperature of 8 °C. Individuals were selected *in situ* on thallus length basis through direct observations and measurements. Thallus age was estimated according to the well-known relationship between age and thallus length reported by *Khailov & Firsov (1976)* for *G. barbata* from the Black Sea, and by *Bianchelli et al. (2023)* for *G. barbata* from Adriatic Sea. Therefore, we identified two groups of algae, each of five individuals: the adults, over one year old, with a thallus length in the range 40–50 cm, and the juveniles, less than one year, with a thallus length within 10–20 cm. Thalli were transported to the laboratory in tanks filled with marine water and maintained at the temperature of almost 8 °C ± 1, which corresponds to water temperature at the sampling time. Photosystem II (PSII) chlorophyll-*a* fluorescence measurements *in vivo* were performed at room temperature of 18 ± 1 °C, taking care to maintain each thallus submerged in sampling water, maintained at almost 8 °C ± 1, to avoid thermal shock to the algae. Before biochemical determinations, thalli were carefully cleaned from any epiphytes and debris with demineralized water. Juveniles and adult thalli were compared for thallus dry matter content (TDMC), photosynthetic pigments, PSII D1 and Rubisco proteins, and antioxidants.

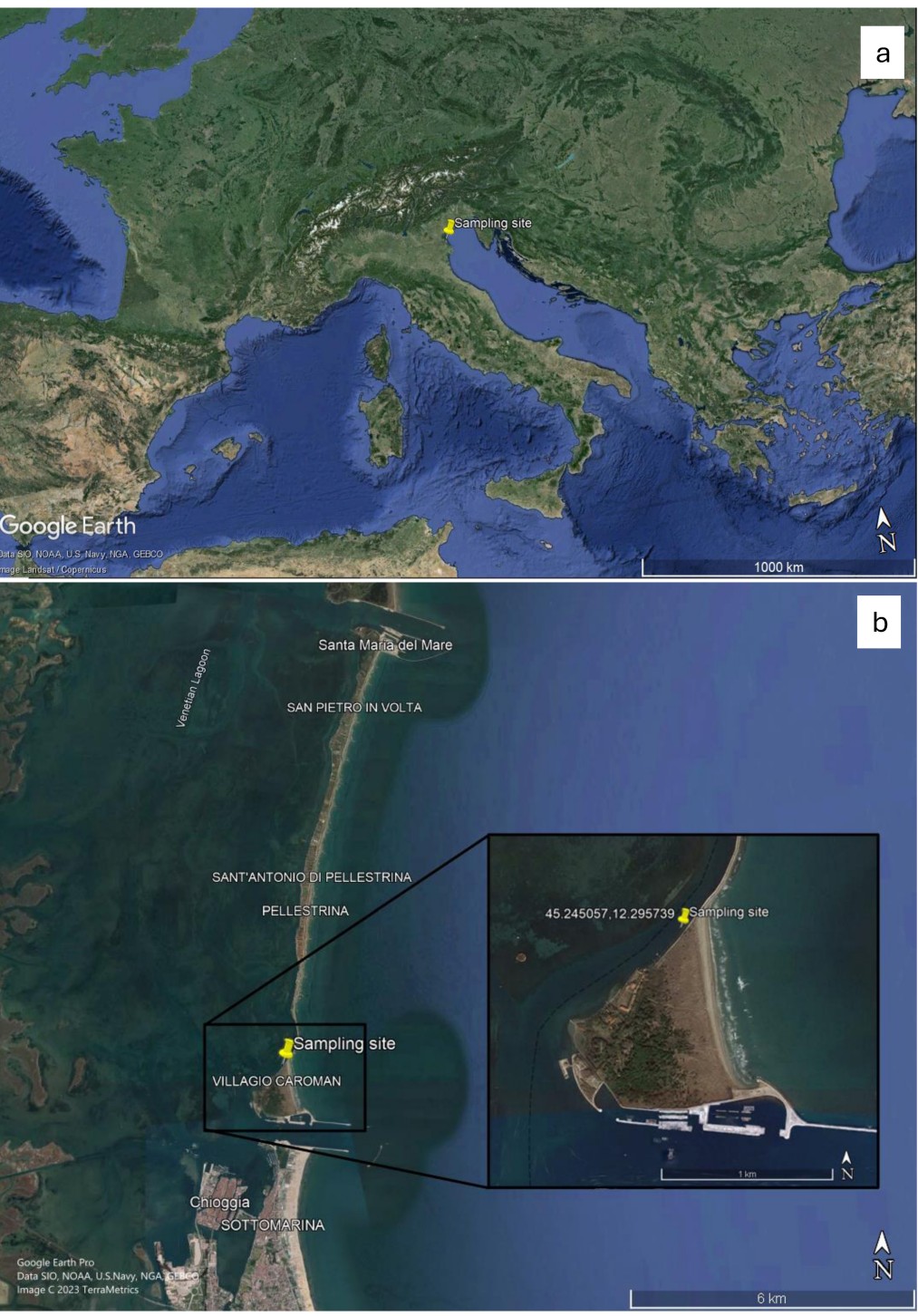

**Figure 1 Images of the sampling site identified by coordinates retrieved from Google Earth Pro Software.** (A) Sampling location identified in European geographical context; (B) sampling site in the natural reserve of Ca' Roman within the Venice lagoon. (continued on next page...)

**Figure 1 (…continued)**
The location is marked by coordinates and displayed from a satellite perspective. Images are retrieved from Google Earth Pro Software (Map data ©2023 Google: Image Landsat Copernicus; Map data ©2023 Google: Image ©2023 Terrametrics).

To characterize the sampling site, we retrieved the environmental variables data from the Copernicus Marine Environment Monitoring Service (*CMEMS*) database (https://marine.copernicus.eu/), using the products Global Ocean Physics Analysis and Forecast and Global Ocean Biogeochemistry Analysis and Forecast (DOI: 10.48670/moi-00016, Accessed on 19-06-2023). We covered the period from October 2020 to February 2023 and selected the geographical coordinates of the sampling site. Data of sea water temperature (T), and water salinity (S) derived from the product Global Ocean Physics Analysis and Forecast (https://data.marine.copernicus.eu/product/GLOBAL_ANALYSISFORECAST_PHY_001_024/description).

This dataset provides gridded data with a spatial resolution of $0.083° \times 0.083°$ ($\sim$9 km $\times$ 9 km) and hourly temporal resolution aggregated with a weakly scale for the plot.

The sea water pH (pH), and molar concentration of nitrate ($NO_3^-$) and phosphate ($PO_4^{3-}$) were derived from the product Global Ocean Biogeochemistry Analysis and Forecast (https://data.marine.copernicus.eu/product/GLOBAL_ANALYSISFORECAST_BGC_001_028/description), and were acquired with a spatial resolution of $0.25° \times 0.25°$ ($\sim$30 km $\times$ 30 km) and hourly temporal resolution aggregated with a weakly scale for the plot. Both datasets derived from the numerical resolution of a global ocean model and are available in the Hindcast format data (an assessment of the past state of the ocean variables made using numerical models with or without data assimilation from satellite and *in situ* observation). For data processing, in both cases, the grid points closer to our target site have been selected.

## Photosynthetic efficiency of thalli

Photosystem II (PSII) chlorophyll-a fluorescence analysis *in vivo* was performed by means of pulse amplitude modulated fluorometer (Junior-PAM, Walz Gmbh, Effeltrich, Germany) on the apical part of juvenile and adult thalli of *G. barbata* to assess the photosynthetic performance. Photosynthetic activity was measured in response to increasing irradiance levels, to assess the light-use efficiency of thalli in photochemistry. Rapid light curves (RLCs) were performed on four individuals for each group ($n = 4$) considering eight actinic light steps at a Photosynthetic Photon Flux Densities (PPFD) of 125, 190, 285, 420, 625, 820, 1,150, 1,500 µmol photons m$^{-2}$ s$^{-1}$ and lasting 60 s each to allow the steady-state fluorescence in actinic light (*Nielsen & Nielsen, 2008*; *Porzio et al., 2020*). Thalli were disposed at 0.5 mm from an optic fibre of 1 mm diameter inclined at 45° respect to samples immersed in seawater suspension. To measure the PSII maximum photochemical efficiency, $F_v/F_m$, thalli were 15 min dark-adapted to allow full oxidation of the PSII reaction centres (*Porzio et al., 2017*). Basal fluorescence ($F_0$) was achieved by applying a weak blue light signal (*Alestra & Schiel, 2015*; *Fabbrizzi et al., 2023*) of 1–2 µmol photons m$^{-2}$ s$^{-1}$, the maximum fluorescence level in the dark ($F_m$) was obtained by applying a saturating light pulse of 7,000 µmol photons m$^{-2}$ s$^{-1}$. The maximal

photochemical efficiency of PSII, $F_v/F_m$, was calculated according to the formula: $F_v/F_m = (F_m\text{-}F_0)/F_m$ (*Beer, Björk & Beardall, 2014*). The quantum yield of the PSII electron transport ($\Phi_{PSII}$) was calculated according to *Genty, Briantais & Baker (1989)* following the equation: $\Phi_{PSII} = (F_m'-F_t)/F_m'$. The Electron Transport Rate (ETR) was evaluated as: ETR $= \Phi_{PSII}*PAR*0.5*A\,F_{II}$ (*Schreiber, 2004*), where the value of 0.5 represents the energy equally distributed between the two photosystems, while A $F_{II}$ is the fraction of light absorbed by Photosystem II, corresponding to the value 0.8 in Phaeophyta (*Celis-Plá et al., 2016*). Non-photochemical quenching (NPQ) was determined as: NPQ $= (F_m - F_m')/F_m'$ (*Bilger & Björkman, 1990*).

## Thallus dry matter content

The thallus dry matter content (TDMC) consists of the proportion of structural compounds and water-filled tissues, which are mainly photosynthetically active. It represents an important functional trait often utilized to assess the adaptability of algae to its environment (*Cappelatti, Mauffrey & Griffin, 2019*). For the TDMC determination, single individuals ($n = 5$) were weighed soon after sampling to determine the fresh mass, successively samples were dried in an oven at 37 °C for 24 h and weighted up to constant dry mass. Finally, the ratio between dry and fresh mass was determined.

## Photosystem II D1 protein and Rubisco determination

Algal samples were fine grounded with liquid nitrogen by a mortar and pestle. Samples were kept on ice in Eppendorf and mechanically homogenized using a pestle and 200 µl of 1x PEB (protein extraction buffer, product no AS08300, Agrisera, Vännäs, Sweeden). Then, samples were centrifuged at 14.000 rpm for 20 min at 4 °C and the supernatants transferred into new tubes. Protein extracts were quantified with the Bradford assay (*Bradford, 1976*), using the BioRad Protein Assay Dye Reagent Concentrate (Bio-Rad Laboratories, Hercules, CA, USA) and the bovine serum albumin (BSA) has been used as a protein standard. The SDS-PAGE (10%) was carried out following *Vitale et al. (2022)* with slight modifications. Briefly, the western blot procedure started with the blocking solution (100 mM Tris-HCl, pH 8.0, 150 mM NaCl, 0.1% Tween20, 5% Milk). To reveal the protein of interest, samples were incubated with the primary antibody (Agrisera, Vännäs, Sweeden) anti-PsbA (rabbit, 1:15,000 v/v, AS05 084) for D1 protein of PSII and anti-RbcL (rabbit, 1:10,000 v/v, AS03037) for Rubisco. Goat anti-Rabbit IgG (H&L), HRP conjugated (1:6,000 v/v, AS09 602) was used as the secondary antibody. Immuno-revelation was performed using the kit for chemiluminescence (Westar supernova, Cyanagen Srl, Bologna, Italy) *via* ChemiDoc System (Bio-Rad, Hercules, CA, USA). The software Image Lab version 5.2.1 (Bio-Rad Laboratories, Hercules, CA, USA) was utilized for the densitometric analysis: band signals were quantified, and the background values were subtracted to obtain and adjusted volume in counts for each band. The density value was expressed in arbitrary units and represented as a boxplot.

## Photosynthetic pigment content analysis

Photosynthetic pigments content, namely total chlorophylls ($a + c$) and total carotenoids, was determined on five individuals per group, considering one individual as one replicate,

and performing three pseudo-replicates per replicate ($n = 15$). The analysis was performed according to *Jeffrey & Humphrey (1975)* and *Lichtenthaler (1987)* following the procedure reported in *Porzio et al. (2017)*. Samples from each thallus (0.040 g of dried powder) were mechanically extracted in 100% acetone inside glass test tubes and left to rest for half an hour in ice and total darkness, to avoid photo-oxidation phenomena. The extracts were centrifuged at 5,000 rpm for 5 min in a Labofuge GL (Heraeus Sepatech, Hanau, Germany). The sample absorbance was measured by a spectrophotometer (UV–VIS Cary 100; Agilent Technologies, Santa Clara, CA, USA) at wavelengths of 662 nm, 630 nm, and 470 nm for chlorophyll a, chlorophyll c and total carotenoids, respectively. Pigment concentration was expressed as $\mu g\ g^{-1}$ of dried weight ($\mu g\ g^{-1}$ DW).

## Soluble antioxidants and antioxidant capacity determination

The polyphenol content was evaluated on five individuals per group, considering one individual as one replicate, and performing three pseudo-replicates each replicate ($n = 15$), through the Folin-Ciocalteu method following the procedure reported in *Fabbrizzi et al. (2023)*. Methanolic extracts were made pestering 0.200 g of dried powder in 2 ml of cold methanol and were stored at 4 °C for 24 h to ultimate the extraction. Then, samples were centrifuged at 4 °C, 11,000 rpm for 10 min in a SL 16R centrifuge (Thermo Fisher Scientific™, Waltham, MA, USA). Then, the supernatant was mixed with 10% Folin–Ciocâlteu solution, 1:1 v/v, and after 3 min, 700 mM $Na_2CO_3$ solution was added to the resulting mixture (1:5, v/v). Samples were incubated for 45 min in total darkness, and the absorbance was measured at 765 nm by a spectrophotometer (UV–VIS Cary 100; Agilent Technologies, Santa Clara, CA, USA). The total polyphenol content was expressed as mg of Gallic Acid Equivalents $g^{-1}$ DW (mg GAE $g^{-1}$ DW) using a gallic acid standard curve. The total flavonoid content was assessed according to the procedure of *Moulehi et al. (2012)* and *Sun, Ricardo-da Silva & Spranger (1998)*. Methanolic extracts were mixed with a solution of 5% $NaNO_2$ (ratio 3:1 v/v); after 6 min, a 10% solution of $AlCl_3$ and a 1M solution of NaOH were added, adjusting the volume with distilled water. Samples were left resting in darkness for 15 min to let the colorimetric reaction happen and finally the absorbance was measured at a wavelength of 510 nm. The total flavonoid content was estimated through a standard catechin curve and expressed as mg of catechin equivalent per gram of dried weight (mg CAT $g^{-1}$ DW). Total condensed tannins were estimated by modifying the procedures described by *Sun, Ricardo-da Silva & Spranger (1998)* and *Moulehi et al. (2012)*, as reported by *Costanzo et al. (2022)*. Briefly, 2.5 mL of methanol-vanillin solution and 2.5 mL of 97% $H_2SO_4$ were mixed with one mL of sample methanolic extract. Then, the mixture was incubated for 15 min in total darkness, and the absorbance was measured at 500 nm. Tannins were quantified with a catechin standard curve and expressed as mg catechin equivalents per gram of dry weight (mg CAT $g^{-1}$ DW). The antioxidant capacity was measured through the DPPH (2,2-diphenyl-1-picrylhydrazyl) assay, where 0.067 mL of methanolic extracts were added to two mL of $6 \times 10^{-5}$ M DPPH in methanol solution and heated at 37 °C for 20 min in a dry bath (Benchmark Scientific, My block™ Mini Dry Bath). Then, absorbance was measured at a wavelength of 515 nm. Antioxidant capacity was

assessed using Trolox as positive control and expressed as percentage of radical inhibition using the formula: % inhibition = ((white Abs − sample Abs)/ white Abs) *100.

## Statistical analysis

To assess the statistically significant differences between the groups (adults and juveniles), the PSII maximal photochemical efficiency ($F_v/F_m$), electron transport rate (ETR), non-photochemical quenching (NPQ), photosynthetic pigments content, photosynthetic protein amounts and antioxidant concentrations were compared performing $t$-test using the Sigma-Stat 12.0 software (Jandel Scientific, USA). Differences were considered statistically significant for $P \leq 0.05$. The Shapiro–Wilk test was applied to check the normality of the data. Whenever the Shapiro–Wilk test failed, the Mann–Whitney Rank Sum Test was applied. The results reported correspond to the average ± standard error. Asterisks were used to indicate statistically significant differences (*** $P \leq 0.001$, ** $P \leq 0.01$, * $P \leq 0.05$, ns $P \geq 0.05$). Boxplots report interquartile range, mean line, whiskers, and outliers. For the RLCs, $t$-tests were performed on the whole data set of RLC-curves and at each PPFD value. All data, including environmental variables, were plotted and visualized by means of R environment software (version 4.2.2., *R Core Team, 2022*) using the *ggplot2* package version 3.5.0.

## RESULTS

### Environmental variables at the sampling site

The environment of the sampled specimens of *G. barbata* is classified as a transitional water system (TWS). It is part of the Venice Lagoon and is closely related to the Adriatic Sea. For these reasons the biogeochemical variables of TWS are strongly influenced by those of the seawater environment, and by human activities (*Solidoro et al., 2010*).

The analysis of biogeochemical variables at the sampling site indicated that seasonal variations of temperature (Fig. 2A) showed peaks of 25–26 °C during summer and of 8–9 °C during winter. The pH fluctuations, reported on the total scale, were in line with the general trend of the Adriatic Sea and ranged from 8.045 to 8.175 (Fig. 2B).

The salinity showed seasonal variations slightly different between 2021 and 2022 spring-summer periods likely due to higher loads of freshwater from estuaries and abundant rainfalls in 2022 more than in 2021 (*ARPAV, 2024*) (Fig. 2C).

Events such as high nutrient loads, mainly nitrogen and phosphorous, and consequent eutrophication often occurred in the Venice Lagoon since the 1920s, along with the direct release of heavy metals and organic micropollutants (*Morand & Briand, 1996*; *Caliceti et al., 2002*; *Pastres et al., 2004*). The wastewater treatment plant processes and the total ban of phosphorous in detergents since 1989, contributed to reduce the nutrient loads (*Acri, Braga & Aubry, 2020*; *Zirino et al., 2016*). In the years 2020–2023, nitrogen loads were below the threshold of 18 $\mu$mol L$^{-1}$ for water bodies with salinity >30 (Fig. 2D) set by the national legislation for the implementation of the *Gazetta Officiale (2011)*, with peaks exceeding the limit only during winter. In contrast, phosphorous levels always exceeded the threshold of 0.48 $\mu$mol L$^{-1}$, except during summer, when usually phytoplankton blooms occurred, and primary productivity was the highest (Fig. 2D). Even if the trophic
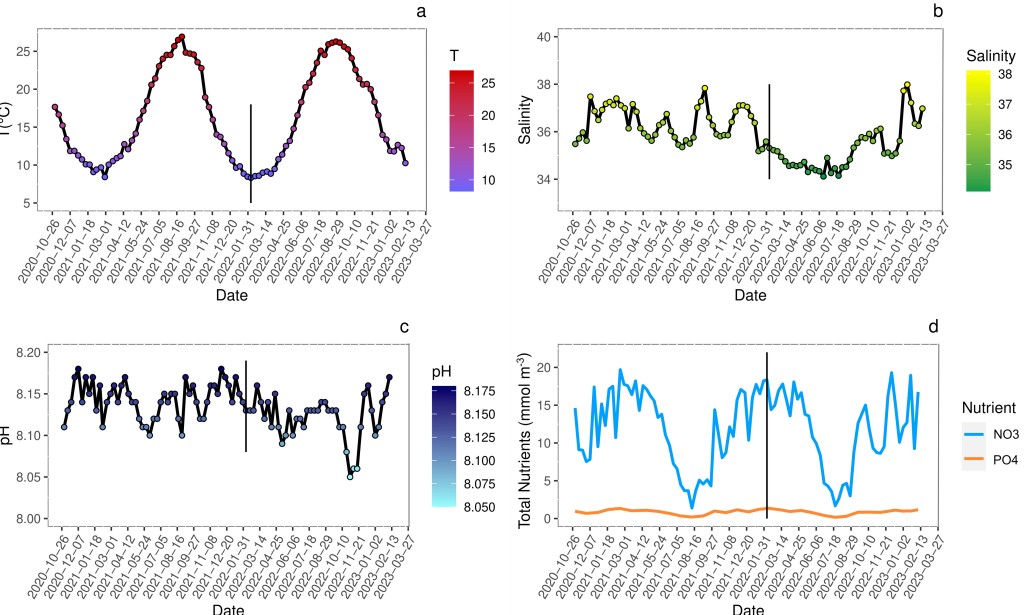

**Figure 2  Seasonal variations (date YYYY/MM/DD) of environmental conditions measured at sampling site.** Analysis of seasonal variations (date YYYY/MM/DD) of environmental conditions measured at sampling site. Straight black line marks the approximate start of juveniles growth. (A) Sea surface temperature (°C); (B) salinity; (C) pH reported on total scale; (D) nitrates ($NO_3$) and phosphates ($PO_4$) concentration (mmol m$^{-3}$). All data are retrieved from Copernicus Marine Environment Monitoring Service (CMEMS) database using the products Global Ocean Physics Analysis and Forecast and Global Ocean Biogeochemistry Analysis and Forecast (DOI: 10.48670/moi-00016, Accessed on 19-06-2023) selecting the geographical coordinates of the sampling site (45°14′42.2″N 12°17′44.7″E).

status of the Venice Lagoon improved significantly (*Çevirgen et al., 2020*), the comparison between years 2017–2019 (*Regione Veneto et al., 2021*) and 2020–2023 (Copernicus Marine Environment Monitoring Service; https://marine.copernicus.eu/) evidenced a significant increase of nitrogen and phosphorous concentrations. To date a discrete environmental state was found utilizing the Trophic Index (TRIX) assessment (*Vollenweider et al., 1998*), for most of the lagoonal waters (*Çevirgen et al., 2020*; *ARPAV TRIX, 2022*).

## Thallus dry matter and photosynthetic pigments content

TDMC showed a significant difference between the groups with 1.3-fold higher values ($P = 0.019$) for adult than juvenile thalli. Pigments differed significantly between adults and juveniles, showing values 1.8-fold higher in adults for total chlorophylls ($P = 0.001$) and 1.7-fold higher for carotenoids ($P = 0.016$) compared to juveniles (Table 1).

## Rapid light curves

The analysis of the quantum yield of PSII electron transport (Fig. 3A) evidenced higher values ($P < 0.05$) in juveniles. Specifically, up to 125 µmol photons m$^{-2}$ s$^{-1}$, the two groups did not show any statistical differences. Conversely, in the range of PPFD from 190 to 1,500 µmol photons m$^{-2}$ s$^{-1}$ juveniles showed higher $\Phi_{PSII}$ values ($P \leq 0.01$) compared to adults. The PSII electron transport rate statistically differs ($P < 0.05$) between adults

**Table 1** **Thallus dry matter content (TDMC) and photosynthetic pigments content in adults and juveniles.** Data are reported as mean $\pm$ SE (TDMC $n = 5$, pigments $n = 15$).

|  | Adults | Juveniles |
|---|---|---|
| TDMC (g g$^{-1}$ DW) | 0.131 $\pm$ 0.005[*] | 0.103 $\pm$ 0.008 |
| Total chlorophylls ($\mu$g g$^{-1}$ DW) | 609.34 $\pm$ 41.25[**] | 340.77 $\pm$ 36.68 |
| Total carotenoids ($\mu$g g$^{-1}$ DW) | 166.17 $\pm$ 17.51[*] | 99.72 $\pm$ 13.09 |

**Notes.**
Asterisks indicate the statistically significant differences ([**]$P \leq 0.01$, [*]$P \leq 0.05$) according to $t$-test.

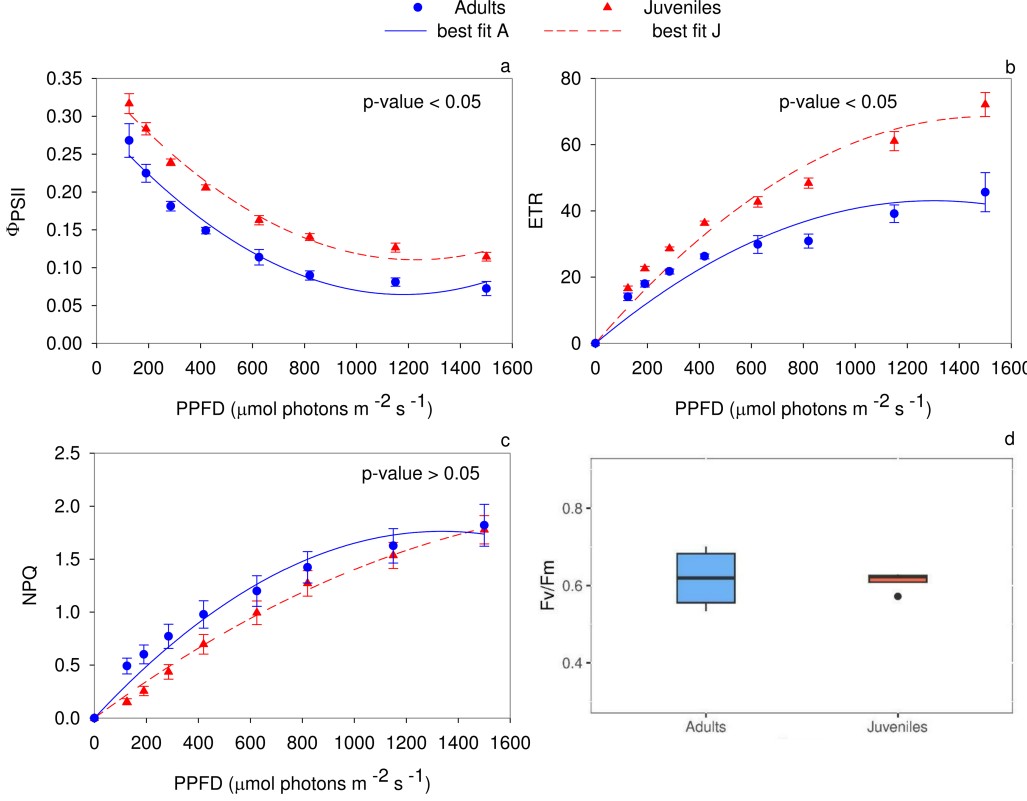

**Figure 3** **Rapid light curves for the photochemical parameters.** RLCs for the photochemical parameters: (A) Quantum yield of PSII electron transport, $\Phi$ PSII; (B) Electron transport rate of PSII (ETR); (C) Non-Photochemical Quenching (NPQ); (D) maximum PSII photochemical efficiency, $F_v/F_m$. Data are reported as means $\pm$ SE ($n = 4$). Statistically significant differences were checked according to $t$-test.

and juveniles. In detail, starting from 285 $\mu$mol photons m$^{-2}$ s$^{-1}$, young thalli, compared to adults, showed a higher electron transport activity ($P \leq 0.01$) (Fig. 3B). Conversely, the non-photochemical quenching (NPQ) exhibited statistically significant differences between young and adult individuals only in the range from 125 to 285 $\mu$mol photons m$^{-2}$ s$^{-1}$, reaching the highest ($P \leq 0.01$) value for adult thalli (Fig. 3C). The maximum photochemical efficiency ($F_v/F_m$) was not statistically different ($P \geq 0.05$) between the two groups with a mean value of 0.619 $\pm$ 0.041 in adults and 0.612 $\pm$ 0.013 in juveniles (Fig. 3D).

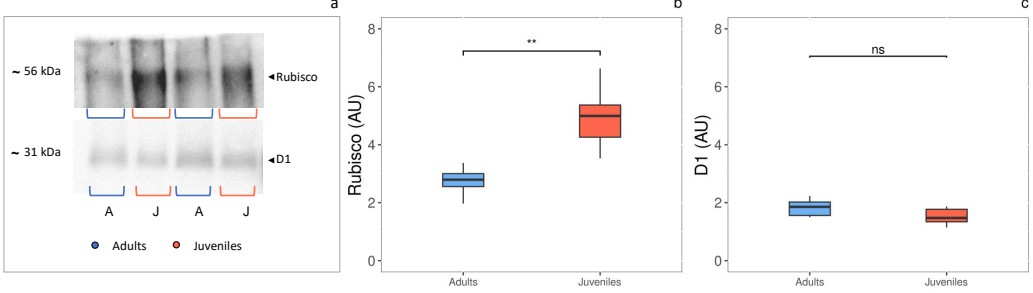

**Figure 4  Western blot and densitometric analysis of Rubisco and D1 proteins.** (A) Western blot of Rubisco and D1 proteins with relative molecular weights; (B) densitometric analysis in arbitrary units of Rubisco protein; (C) densitometric analysis in arbitrary units of D1 protein. Asterisks indicate the statistically significant differences (***$P \leq 0.001$, **$P \leq 0.01$, *$P \leq 0.05$, ns $P \geq 0.05$) according to $t$-test. The images of western blot were downloaded by ChemiDoc System (Bio-Rad).

## Photosystem II D1 protein and Rubisco

The densitometric analysis of the D1 protein (Figs. 4A, 4C) showed no statistically significant difference between adults and juveniles. Conversely, the Rubisco protein amount was 1.8-fold higher in juveniles ($P = 0.001$) than in adults, exhibiting an increase of 44% (Figs. 3A, 3B).

## Soluble antioxidants content

Total polyphenols content showed significant difference ($P = 0.0003$) in response to thallus age, with values 1.4-fold higher in juveniles ($4.948 \pm 0.206$ mg GAE g$^{-1}$ DW) than in adults ($3.570 \pm 0.102$ mg GAE g$^{-1}$ DW) (Fig. 5A). The same behavior was observed for total flavonoids, whose concentration was 2.0-times higher ($P < 0.00001$) in juveniles than in adults, with values of $13.373 \pm 0.662$ and $6.554 \pm 0.279$ mg CAT g$^{-1}$ DW, respectively (Fig. 5B). Also, tannins were 1.36-fold higher ($P = 0.012$) in juveniles ($74.880 \pm 4.387$ mg CAT g$^{-1}$ DW) than in adults ($55.075 \pm 4.345$ mg CAT g$^{-1}$ DW) (Fig. 5C). Finally, the radical scavenging activity was 1.2-times higher ($P < 0.00001$) in juveniles (% inhibition $= 73.463 \pm 0.331$) than in adults (% inhibition $= 61.588 \pm 1.164$) (Fig. 5D).

## DISCUSSION

Our study explored if *G. barbata* is featured by specific functional, eco-physiological and biochemical traits, allowing an efficient use of habitat resources, and if this potential might change with individual growth. The results evidenced that juveniles and adults differed in their photophysiological traits, modulation of antioxidant production and biomass partitioning into photosynthetic and non-photosynthetic tissues.

Our study area, experiences seasonal fluctuations in salinity, temperature, and nutrients. These fluctuations are within the tolerance range documented for the *G. barbata* species (*Orfanidis, 1991*; *Irving et al., 2009*; *Iveša et al., 2022*), which is found in euhaline and polyhaline environments (*Sadogurska et al., 2021*; *Tursi et al., 2023*). The species seems potentially also tolerant to pH decrease (*Celis-Plá et al., 2017*).

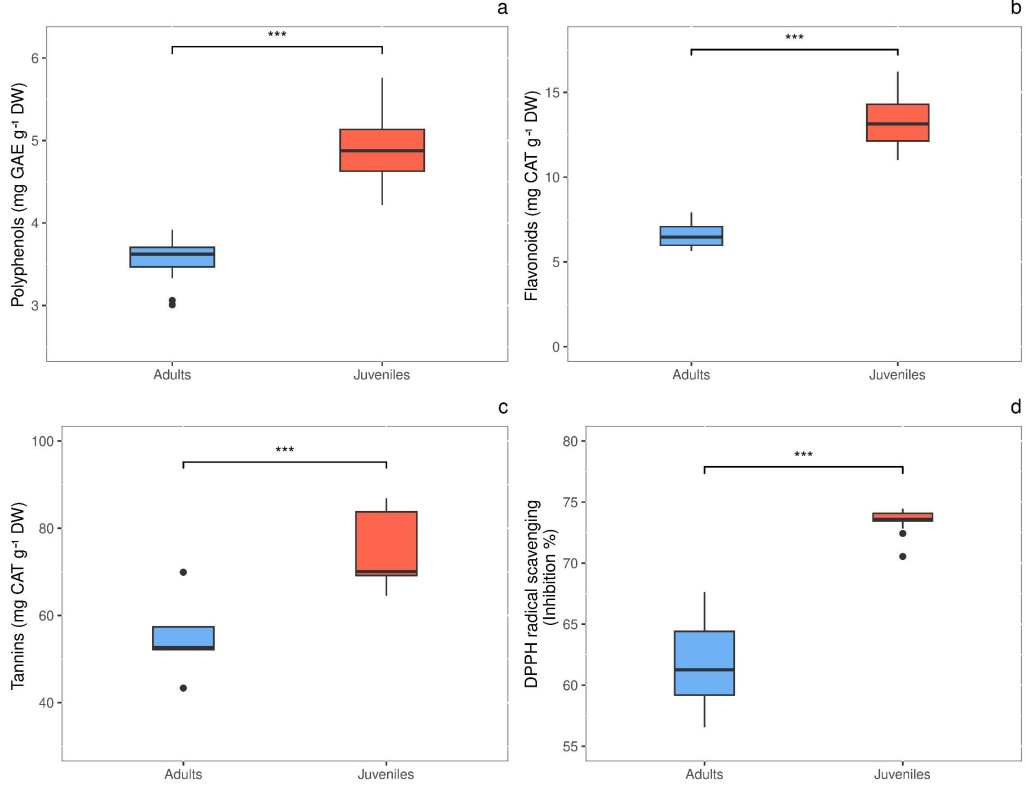

**Figure 5** **Antioxidant content in adult and juvenile *G. barbata* thalli.** (A) Total polyphenols; (B) flavonoids; (C) tannins; (D) antioxidant capacity, measures as radical scavenging activity. Data are reported as means $\pm$ SE ($n = 15$). Asterisks indicate the statistically significant differences (***$P \leq 0.001$, **$P \leq 0.01$, *$P \leq 0.05$) according to *t*-test.

*Gongolaria barbata* is also well adapted to unusually high as cold temperatures occurring throughout the northern Adriatic Sea (*Iveša & Devescovi, 2014*; *Iveša, Djakovac & Devescovi, 2016*; *Iveša et al., 2022*). Not only adults in the vegetative phase can overcame temperature up to 30–34 °C (*Iveša et al., 2022*), but also recruits can trigger physiological acclimation mechanisms to cope with marine heat waves (*Fabbrizzi et al., 2023*). During winter, eight and even two-month-old recruits survived at temperatures close to or below 'zero' without physical damage (*Orfanidis, 1991*; *Iveša et al., 2022*). The temperatures recorded during the period 2020–2023 fall within the temperature tolerance of the species (*Orfanidis, 1991*), well adapted to these specific environmental conditions and likely capable of withstanding extreme winter and summer events. Several studies (*Iveša & Devescovi, 2014*; *Iveša, Djakovac & Devescovi, 2016*; *Iveša et al., 2022*) also evidenced that populations of *G. barbata* inhabit lagoons, systems with not-limiting nutrient patterns, located in the northern Adriatic Sea. In sites such as the Venice Lagoon, the availability of nutrients and irradiance are driving factors in regulating the photosynthetic activity and growth of macroalgae. During thalli development, resource requirements and management may change depending on seasonal environmental fluctuations and individual age (*Harrison & Hurd, 2001*;

*Taylor, Fletcher & Raven, 2001*; *Delgado, Ballesteros & Vidal, 1994*; *Stengel & Dring, 1998*; *Stengel, McGrath & Morrison, 2005*). The absorption of micronutrients, such as trace metals (copper, iron and manganese), is indispensable for several metabolic functions, like enzyme activation, photosynthetic electron transport and nitrogen metabolism. Their availability is regulated by water temperature, pH and salinity and accumulation rate in brown algae may differ depending on thallus age and species (*Stengel, McGrath & Morrison, 2005*). Macronutrients as nitrogen (N) and phosphorous (P) play a pivotal role in the regulation of photosynthesis and growth rate of seaweeds. In the investigation area, N and P in available forms of nitrates and phosphates, exhibited the lowest concentration during summer and the highest levels during winter. In particular, in winter both adult and juvenile thalli may accumulate nutrients as a reservoir for next periods of nutrient limitation (*Harrison & Hurd, 2001*; *Delgado, Ballesteros & Vidal, 1994*; *Celis-Plá et al., 2016*). N and P-storage capacity and utilization varied between juvenile and adult stages of *G. barbata* (*Delgado, Ballesteros & Vidal, 1994*; *Harrison & Hurd, 2001*; *Ohtake et al., 2021*). Generally, young tissues uptake more nutrients than older (*Delgado, Ballesteros & Vidal, 1994*; *Harrison & Hurd, 2001*; *Ohtake et al., 2021*) indicating a higher investment of N and P in the photochemical and assimilation reactions thus supporting the highest photosynthetic activity found in juveniles compared to adults.

However, regardless of the age, our results indicate no stress occurrence for photosynthetic apparatus and a fully functional PSII photochemistry in both juveniles and adult thalli. Indeed, we observed in both groups an $F_v/F_m$ ratio close to 0.7 (*Baker, 2008*), consistent to values found in other *Cystoseira spp.* (*Celis-Plá et al., 2016*; *Mancuso et al., 2019*). *Kaleb et al. (2023)* demonstrated that juvenile thalli of *G. barbata* from sites close to Marano and Grado lagoon (Adriatic Sea), benefit from the nutrient enrichment of the habitat even at temperatures of 10 and 14 °C, suggesting that also in our case, *G. barbata* thalli irrespective of the age, manage well local available nutrients.

Consistent with no difference in $F_v/F_m$ ratio, PSII D1 protein amount did not vary between young and adult thalli, highlighting that the photosynthetic efficiency reduction found in adults was due to a down-regulation of photochemistry rather than to an impairment of photosystems. It is well known that the D1 protein is the primary target of light-induced oxidative damages (*Mulo, Sakurai & Aro, 2012*), and that its decrease is often correlated with photoinhibition and oxidative stress of PSII, resulting in a reduction of $F_v/F_m$ ratio (*Schofield, Evens & Millie, 1998*). Our results support the evidence that photosynthetic apparatus of both juveniles and adults perform, and the high photochemical efficiency is pivotal in providing the adaptive solid potential of *G. barbata* thalli in its habitat.

Conversely to $F_v/F_m$ ratio, quantum yield of PSII electron rate ($\Phi_{PSII}$), electron transport rate (ETR), non-photochemical quenching (NPQ), as well as pigment and protein production were differently modulated in juveniles and adults suggesting that young individuals improve photosynthesis by allocating more nutrients into components of the electron transport chain and Rubisco synthesis, while the adult ones utilize nitrogen mainly to potentiate the pigment content synthesis (*Harrison & Hurd, 2001*; *Gómez, Wiencke & Thomas, 1996*; *Stengel & Dring, 1998*; *Ohtake et al., 2021*).

According to the higher photosynthetic efficiency found in younger than older individuals (*Gómez, Wiencke & Thomas, 1996*; *Kim & Garbary, 2009*), the rapid light curves evidenced how juveniles and adults differently used harvested light in the photosynthetic process at both unsaturated and saturated irradiances. More specifically, juveniles invest more light energy into photosynthesis, displaying 1.6-fold higher quantum yield of PSII electron transport ($\Phi_{PSII}$) and electron transport rate (ETR), along with 1.8-fold higher content of Rubisco. It is likely that young individuals invest more of the available P in ATP energy transfer during photosynthesis (*Xu et al., 2017*; *Ohtake et al., 2021*). This is because in young thalli, the higher activity of the electron transport chain is expected to produce more ATP and NADPH molecules, which are used in $CO_2$ fixation by Rubisco. Rubisco concentration increases with high photochemical activity (*Raven, 1997*; *Gylle et al., 2013*). Our data align with other studies showing that the amount of Rubisco may decrease as thallus age increases. The age-related Rubisco and photosynthesis reduction is consistent with other studies on higher plants (*Gómez, Wiencke & Thomas, 1996*; *Bertamini & Nedunchezhian, 2002*).

As a result of a compensation mechanism, the higher investment of light energy in photochemical activity observed in young thalli implies the reduction of dissipation processes, explaining the lower NPQ values. Conversely, adult thalli, more exposed to unfavorable water surface conditions (*i.e.,* light excess during tide events, higher UV levels, high or low temperature) than young, submerged thalli, showed higher NPQ values in the PPFD range from 100 to 500 $\mu$mol photons $m^{-2}$ $s^{-1}$, dissipating the excess light mainly as heat.

The higher levels of carotenoids in adult thalli, together to the NPQ rise (*Lavaud & Goss, 2014*), may suggest the activation of thermal dissipation mechanisms mediated by xanthophyll cycle for photoprotection purposes (*Demmig-Adams & Adams III, 2006*; *Celis-Plá et al., 2016*).

Beyond a photoprotective role, the higher photosynthetic pigment content found in adults (1.8 and 1.7 folds for chlorophylls and carotenoids, respectively) compared to juveniles may be also a way to compensate for the reduced photochemistry (*Porzio et al., 2017*). However, it cannot be excluded that the increase of chlorophyll *a* with age during winter, may be also due to the combination of low surface irradiances and shading produced by other individuals or self-shading linked to the thallus density and morphology (*Stengel & Dring, 1998*).

As regards the morpho-functional traits, thallus dry matter content (TDMC) significantly differed between young and mature individuals indicating a dissimilar distribution between structural compounds and water-filled and nutrient-rich photosynthetically active tissues (*Cappelatti, Mauffrey & Griffin, 2019*). TDMC modulation during growth is involved in organism survival (*Elger & Willby, 2003*) and its increase is generally associated with high resistance to wave damage and desiccation (*Cappelatti, Mauffrey & Griffin, 2019*).

*Cystoseira s.l.* is known to exhibit seasonal variations in TDMC during its development. This occurs as the thalli shift from primary growth in winter and spring to dormancy in summer and autumn. During dormancy, individuals shed many secondary branches, leading to lower water content (*Orfanidis et al., 2017*; *Iveša et al., 2022*).

In our study, even if TDMC was not measured seasonally, the increase of dry biomass found in adults compared to juveniles during winter, when temperature are low, suggests an investment of carbon in structural compounds with age. This response might provide an advantage for adult individuals coping with waves or grazing pressure. A positive correlation exists between thallus-size, life-stage and wave action (*Thomsen, Wernberg & Kendrick, 2004*). Considering the different size between juvenile and adult individuals of *G. barbata*, wave-generated forces may represent an environmental stimulus that elicits changes in the carbon metabolism, inducing adaptive variations in morphology, and structural and biomechanical properties of adult thalli. Such changes may consist in the dislodgement of carbohydrate polymers toward cell wall synthesis (*Kraemer & Chapman, 1991*; *Dudgeon & Johnson, 1992*), making adults more resistant and attached to the substrate than juveniles (*Kraemer & Chapman, 1991*; *Cappelatti, Mauffrey & Griffin, 2019*). The elevated dry matter content of macroalgae is not only an indicator of mechanical resistance but also a proxy to predict variations in palatability for grazers (*Elger & Willby, 2003*). Notably, the difference in TDMC between young and adult thalli was accompanied by a diverse regulation of photosynthetic activity. The reduced palatability and the lower photosynthetic rate suggest that adults prefer a resource conservation strategy and an investment into structural defenses (*Elger & Willby, 2003*; *Mauffrey, Cappelatti & Griffin, 2020*).

The availability of nitrogen in the growth environment and increased photosynthetic activity may explain why juveniles exhibited higher levels of antioxidants compared to adults. Nitrate enrichment has been reported to enhance the accumulation of phenolic compounds in *C. tamariscifolia* (*Celis-Plá et al., 2014*). There is a positive relationship between high photosynthetic rates and the accumulation of carbon with secondary metabolism in the form of phenolic compounds (*Celis-Plá et al., 2016*). The increased synthesis of such scavengers provides photoprotection for photosynthetic apparatus and an advantage against predation.

The photoprotection has been observed in *C. tamariscifolia* thalli habiting in oligo and ultra-oligotrophic transparent coastal waters where higher irradiance levels can be found and photodamage should be prevented (*Celis-Plá et al., 2017*). As this is not the case of the Venice lagoon, we hypothesize that young thalli promote secondary metabolism and antioxidant compounds as a defense against grazing (*Mannino et al., 2016*). Phenolics are particularly abundant in brown macroalgae (*Phaeophyceae*) due to their exclusive production of phlorotannins (*Montero et al., 2019*) and play structural, antibacterial, photoprotective, and herbivore deterrent roles (*Li et al., 2011*; *Steevensz et al., 2012*; *Stiger et al., 2014*; *Mancuso et al., 2019*).

The antioxidant level found in our study is comparable with that reported by other authors who analyzed specimens of *G. barbata* from natural environments (*Cadar et al., 2019*; *Castillo et al., 2023*). We observed higher concentrations of total phenolic compounds in juveniles supporting the hypothesis of the increase of resources allocation into chemical defenses.

Flavonoids are another class of phenolic compounds whose role, even if not thoroughly investigated in algae yet, is primarily to guarantee photoprotection by scavenging
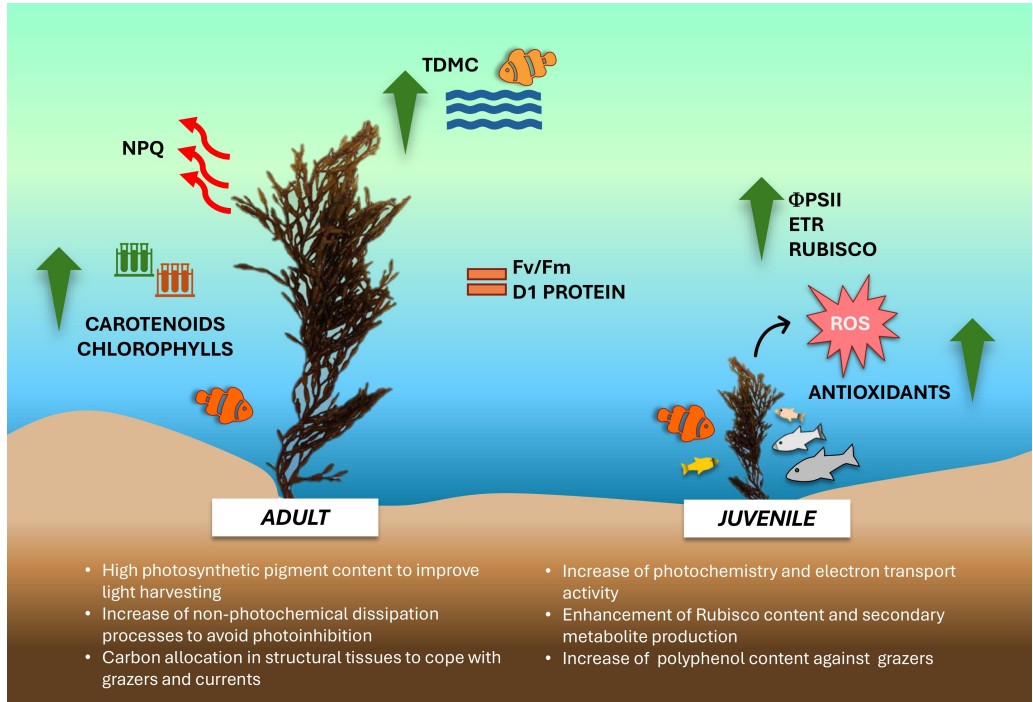

**Figure 6** **Schema of main results obtained in the study.** Conceptual diagram summarizing the conclusions of the study. Image icons from Powerpoint.

reactive oxygen species (ROS) (*Fernando, Lee & Ahn, 2022*). Tannins, involved in cell-wall hardening, exert a structural and protective purpose and, as polyphenols, provide a chemical defense against grazers (*Mannino & Micheli, 2020*). To support the general trend of antioxidant compounds, we found in juveniles a higher radical scavenging activity than in adults. It may be hypothesized that the elevated scavenging activity of young thalli was mainly due to tannins, polyphenols, and flavonoids, as observed in previous studies on other Phaeophyceae (*Connan et al., 2006*; *Nunes et al., 2021*; *Ruiz-Medina, Sansón & González-Rodríguez, 2022*).

## CONCLUSIONS

The overall data suggests that during the winter season, young and adult thalli of *G. barbata* inhabiting the transitional water system of the Venice lagoon, adopt different growth strategies and show remarkable variations in their eco-physiological and biochemical traits (Fig. 6).

Juvenile thalli preferentially utilize the available nutrients, potentiating the photosynthetic components involved in electron transport chain and Rubisco synthesis, thus determining a higher photosynthetic activity and production of secondary metabolites. The elevated production of flavonoids, tannins and polyphenols confers more resistance against potential grazers to juvenile thalli. Conversely, adult thalli, more exposed to unfavorable water surface conditions (*i.e.,* light excess during tide events, higher UV levels,

higher or lower temperature, shading phenomena), show a higher photosynthetic pigment content to compensate for the lower photochemistry and potentiate the thermal dissipation processes as a photoprotective mechanism. Furthermore, being more wave-exposed, adults also exhibit an increase of biomass allocation towards structural tissues, conferring thalli resistance against grazers and wave-generated forces. Despite further studies with an increased samples size in space and time are needed to corroborate our results, these preliminary findings propose 'thallus age' as a valuable and potential ecological trait to assess growth-defense strategies exploited by *G. barbata* during different seasons and against multiple environmental stressors.

### Funding
This research received funding by National Biodiversity Future Center—NBFC, project code MUR: CN00000033—CUP UNINA: E63C22000990007. The funders had no role in study design, data collection and analysis, decision to publish, or preparation of the manuscript.

### Grant Disclosures
The following grant information was disclosed by the authors:
National Biodiversity Future Center—NBFC: MUR: CN00000033, CUP UNINA: E63C22000990007.

### Competing Interests
Carmen Arena is an Academic Editor for PeerJ.

### Author Contributions

- Maria Luisa Pica performed the experiments, analyzed the data, prepared figures and/or tables, authored or reviewed drafts of the article, and approved the final draft.
- Ermenegilda Vitale performed the experiments, analyzed the data, prepared figures and/or tables, authored or reviewed drafts of the article, and approved the final draft.
- Rosa Donadio performed the experiments, analyzed the data, authored or reviewed drafts of the article, and approved the final draft.
- Giulia Costanzo performed the experiments, analyzed the data, authored or reviewed drafts of the article, and approved the final draft.
- Marco Munari performed the experiments, analyzed the data, authored or reviewed drafts of the article, and approved the final draft.
- Erika Fabbrizzi analyzed the data, authored or reviewed drafts of the article, and approved the final draft.
- Simonetta Fraschetti conceived and designed the experiments, authored or reviewed drafts of the article, contributed reagents, materials, and approved the final draft.
- Carmen Arena conceived and designed the experiments, authored or reviewed drafts of the article, contributed reagents, materials, and approved the final draft.

## Field Study Permissions

The following information was supplied relating to field study approvals (i.e., approving body and any reference numbers):

Field experiments were authorised by Regione del Veneto with decree n. 369 (date 04.05.2023).

## Data Availability

The raw measurements are available in the Supplementary Files.

## Supplemental Information

Supplemental information for this article can be found online at http://dx.doi.org/10.7717/peerj.17959#supplemental-information.

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
