# Peer review of "Functional ecological traits in young and adult thalli of canopy-forming brown macroalga Gongolaria barbata (Phaeophyta) from a transitional water system"

_PeerJ, doi:10.7717/peerj.17959_

## Round 0.1 · original submission · Major Revisions

We have received evaluations from three expert reviewers and these can be seen below as well as in an attached PDF. As you can see the evaluations are very thorough, with some concerns and suggestions to improve the manuscript. Please ensure that you respond to all of the issues raised in a rebutttal that includes the changes that were made and where they can be found within the manuscript.

Reviewer 1 ·

Basic reporting

The manuscript entitled “Structural and functional ecological traits in young and adult thalli of canopy-forming brown macroalga Gongolaria barbata (Phaeophyta) from a transitional water system” by Pica ML et al. submitted to PeerJ-Life and Environment presents an original research on the comparison of dry matter content, photosynthetic activity, photosynthetic pigments, and antioxidant capacity between adults and juveniles of Gongolaria barbata and how this may have implications on their biological performance.
This study contributes to the knowledge on the physiological and biological traits of the canopy-forming species, that are so ecologically important but are in regression, especially in the Mediterranean basin. In addition, the authors use different methodologies to obtain different biological variables, offering a wide range of approaches. Therefore, I considered this study to be relevant for publication. However, I have some concerns that I would like to comment on: To characterize the different variables of each growth stage the authors only analyse 5 individuals for each stage from a single population and at a single time of the year (line 102). It is hard to believe that this reduced number of individuals sampled is representative of the whole species.
On the other hand, it seems excessive to refer to photosynthetic yield, antioxidant defences and dry matter content as ecological traits when these are more physiological and/or biological traits. I would recommend tone down the statements arguing in this sense in the introduction and discussion sections (for example in lines 78, 94, 322-323).

Experimental design

As I mentioned above, my mine concern is about the small number of replications. To characterize the different variables of each growth stage the authors only analyse 5 individuals for each stage from a single population and at a single time of the year (line 102). It is hard to believe that this reduced number of individuals sampled is representative of the whole species.

Validity of the findings

The results obtained from the different approaches suggest reasonable and well-argued conclusions, where juvenile individuals present a biology different from that of adults, which is related to their need to be able to grow and protect themselves (juveniles) versus maintenance and resistance (adults).The statistics are well applied and the results are conclusive, although including samples from other populations or other seasons would make the results more robust.

Additional comments

A part, some specific comments:
Line 70. Add a space in “Molinari-Novoa &Guiry”
Line 103. Why was the sampling done in February? Given that these algae have a seasonal phenology, wouldn't it be interesting to carry out seasonal sampling to check that the February results are consistent throughout the year?
Lines 109-111. The mentioned study of Khailov & Firsov, 1976 on the well-established relationship between the thallus length and the age of an individual is carried out in a G. barbata population from the Black Sea, which has particular conditions such as a high nutrients content. This length-age relationship does not seem to be applicable to the population sampled from the Adriatic, as the environmental conditions are probably very different from those in the Black Sea, and therefore the growth rate of this species could also be very different. Could the authors provide some evidence that the growth of the two populations is similar? if not, my suggestion is not to talk about the age of the specimens.
Line 118-161: The authors describe the environmental conditions of the sampled site in detail, but the discussion of these specific conditions in relation to their biological results is limited. I recommend further discussion of these results in relation to environmental conditions.
Line 194. Change “incubated” by “dried”
Line 280. “(P=016)” probably is wrong
Line 291. Remove the point in between brackets “(P≤0.01). (Fig. 3b)”.
Line 320. Ecological traits again
Lines 332-333. Please rewrite the sentence, I think it is not clear the meaning
Line 358: Italicise Cystoseira
Line 416: Add an space in “G.barbata”

·

Basic reporting

The work of Pica and colleagues explores the differences in the photophysiology and biochemical composition between young and adult thalli of the canopy-forming brown seaweed Gongolaria barbata from a transitional water system (Venice Lagoon). The article addresses fundamental aspects on the species’ physiological traits, which provides valuable ecophysiological information to assess the species’ resilience to environmental fluctuations (light) at population level. The overall importance of this study lies on addressing how ecological strategies can vary at intraspecific level (i.e. with individual age, young vs. adult thalli). I find this approach very enriching as it provides a broader view on the species autoecology and has the potential application of designing better conservation strategies.
Overall, the article is easy to follow and read, but it could be further improved on its structure, style and English usage. Specific comments in this line are indicated in the reviewed PDF version with multiple suggestions for improvement. I would encourage the authors to use a more direct style by making use of the active voice, which will help them to convey their messages much clearer.
Literature references are well selected, but some key references might be missing. These are specified in the reviewed PDF along the text, and some suggestions in the general comments section in relation to age-related physiological changes in seaweeds. The field background/context is sufficiently provided in the methods’ section.
The figures and tables are presented in a professional way. In the discussion, I think a conceptual diagram with the main conclusions in relation to energy allocation from young to adults would be advantageous and more illustrative. The raw data would be improved by specifying units for the variables measured in all tabs.
Despite the results are relevant to the hypotheses, the article might need more self-criticism by recognizing its limitations. The arguments for this are elaborated in the general comments section

Experimental design

The study lies within the scope of the journal. The research question is well defined and it has not been addressed before for this species. I consider this is a timely publication given the need for conservation of key canopy-forming Cystoseira sensu lato species in the Mediterranean.
The study is well-designed following high technical an ethical standars (the authors even provided the permits for accessing the sampling area). Nonetheless, the results would be mor robust if the same measurements were taken at different seasons as done for other Cystoseira species (Celís-Plá et al. 2016). If possible, I would encourage the authors to do that to avoid generalization from one-time measurements. Still, it can be understood that it might not be representative to include additional seasons from different years, due to changes in their environmental histories. In that case, the main efforts for the article to be eligible for acceptance should be to acknowledge critically the study limitations.
On the other hand, although it is essential to analyze the environmental context, the data do not relate directly to physiological measures. This information depicts the environmental history of the thalli, but I sense it does not add key information linked to the main results (photophysiology, biochemical composition). Overall, I think this part should be moved to results or even better, I would recommend including it as supplementary material, as it diverts the reader from the main focus of the research.
Furthermore, in relation to the environmental fluctuations fo the transitional water system, some of the variables might show a higher variability at small (daily) scale than seasonal scale, particularly in relation to nutrients. This is mostly due to nutrient fluxes from sediments, benthic fauna or anthropogenic sources (as stated in the article), that can also greatly be modulated by tidal forces. I think these aspects should be considered. Also, it might be helpful to provide the spatial/temporal resolution of the data retrieved and the source (in situ stations? Oceanographic surveys?).
Finally, there is missing information that must be specified, in relation to the conditions at which the thalli were transported to the laboratory, if the thalli were cleaned from epiphytes and at which conditions the PAM measurements were performed (temperature).
Celis-Plá, P. S., Bouzon, Z. L., Hall-Spencer, J. M., Schmidt, E. C., Korbee, N., & Figueroa, F. L. (2016). Seasonal biochemical and photophysiological responses in the intertidal macroalga Cystoseira tamariscifolia (Ochrophyta). Marine Environmental Research, 115, 89-97.

Validity of the findings

The findings show robust analyses following the methods specified in the statistical analysis section.

Despite the valuable approach and validity of the design and results, I find the article very ambitious in its conclusions. I think the work needs more self-criticism by recognizing its limitations, particularly in the discussion. I encourage the authors to reflect on the following aspects:
1) The measurements they perform on G. barbata are photophysiological ones (PAM fluorometry) and biochemical composition (thallus dry matter, pigments, antioxidants, PSII D1 and Rubisco proteins). Following the comments in the reviewed PDF, I encourage the authors to revise the "trait" concepts and the nomenclature used.
2) The measurements were carried out in only one season and directly on field-collected specimens. This implies that:
a. Seasonal variability is not being analyzed, ignoring the possibility that physiological response varies differently between adults and juveniles. For example, due to their different size/architecture of thalli in their natural habitat, microhabitat conditions may differ between young/adult thalli. Since the article analyze light responses, it could be hypothesized that adult thalli experience higher self-shading. What if the changes during the growth period to younger to adults change the microhabitat and induces changes on its energy allocation due to acclimation processes rather than and adaptative features? Using the term “adaptation” would imply long-term evolutionary processes that would explain such differences. Also, light exposure during emersion can be higher for adults (if remain floating in the surface) than for young thalli that might remain submerged.
b. It is not considered neither discussed how investment in reproductive output may be interfering/explaining the changes in adults’ energy allocation. Including phenological data would help to interpret better the results.
3) Since no manipulative laboratory experiments are carried out to assess responses/acclimation to different environmental stressors, I find highly speculative to draw conclusions about the species resilience to different environmental stressors, particularly temperature.

Additional comments

I would encourage the authors to revise classic and recent works on seaweed ecophysiology that report and discuss age-related physiological and biochemical differences. I think it might help them to discuss their results and focus their discussion on the reported differences rather than relate them with the overall species resilience to environmental factors. Here I provide some that may serve them as reference:
Harrison, P. J., & Hurd, C. L. (2001). Nutrient physiology of seaweeds: application of concepts to aquaculture. Cah Biol Mar, 42(1-2), 71-82.
Wang, Y., Xu, D., Fan, X., Zhang, X., Ye, N., Wang, W., ... & Cao, S. (2013). Variation of photosynthetic performance, nutrient uptake, and elemental composition of different generations and different thallus parts of Saccharina japonica. Journal of applied phycology, 25, 631-637.
Gómez, I., Wiencke, C., & Thomas, D. N. (1996). Variations in photosynthetic characteristics of the Antarctic marine brown alga Ascoseira mirabilis in relation to thallus age and size. European Journal of Phycology, 31(2), 167-172.
Stengel, D. B., McGrath, H., & Morrison, L. J. (2005). Tissue Cu, Fe and Mn concentrations in different-aged and different functional thallus regions of three brown algae from western Ireland. Estuarine, Coastal and Shelf Science, 65(4), 687-696.
Kraemer, G. P., & Chapman, D. J. (1991). Effects of tensile force and nutrient availability on carbon uptake and cell wall synthesis in blades of juvenile Egregia menziesii (Turn.) Aresch.(Phaeophyta). Journal of experimental marine biology and ecology, 149(2), 267-277.
Stengel, D. B., & Dring, M. J. (1998). Seasonal variation in the pigment content and photosynthesis of different thallus regions of Ascophyllum nodosum (Fucales, Phaeophyta) in relation to position in the canopy. Phycologia, 37(4), 259-268.
Nunes, D., André, R., Ressaissi, A., Duarte, B., Melo, R., & Serralheiro, M. L. (2021). Influence of Gender and Age of Brown Seaweed (Fucus vesiculosus) on Biochemical Activities of Its Aqueous Extracts. Foods, 11(1), 39.
Connan, S., Delisle, F., Deslandes, E., & Ar Gall, E. (2006). Intra-thallus phlorotannin content and antioxidant activity in Phaeophyceae of temperate waters.
Kim, K. Y., & Garbary, D. J. (2009). Form, function and longevity in fucoid thalli: chlorophyll a fluorescence differentiation of Ascophyllum nodosum, Fucus vesiculosus and F. distichus (Phaeophyceae). Algae, 24(2), 93-104.

·

Basic reporting

A very interesting study! Please find my comments on the PDF file.

Experimental design

One sampling effort but several not commonly assessed parameters have been investigated, e.g., RUBISCO, D1, antioxidants.

Validity of the findings

Very interesting results! Please find my comments on the PDF file.

Additional comments

No

---

## Round 0.2 · Minor Revisions

We have concluded with the evaluation of your manuscript and the comments of one reviewer can be seen below. You have done a great job in addressing the reviewer´s comments, however there are some minor issues that have been pointed out that need to be attended to in a revision.

Please ensure that you answer the concerns point by point in your response to the review.

·

Basic reporting

The authors have addressed all reviewers suggestions and comments, and discussed the rebuttals with a high standard of scientific quality. They have found the balance between highlighting the core results, and recognizing the study limitations in a coherent way. The inclusion of the final conceptual diagram is a plus that elevates the relevance of the conclusions and makes them clearer to the reader. Therefore, I consider they did a strong effort to improve the manuscript.

Experimental design

The methods section has substantially been modified and improved, being sound and accurate

Validity of the findings

The author have cautiously revised all the details, even in the supplementary data. The findings are robust, statistically sound within the sampling limits of the study highlighted in the response to the reviewers.

Additional comments

Hereby I provide some minor comments prior to a final version if accepted:
L76: First time using Cystoseira s.l. do not include directly the abbreviation bu Cystoseira sensu lato (s.l.)
L102: Which paragraph do this sentence belong? It seems isolated there
L110: Rewrite for clarity and correct use of English tense: "However, while adults have been extensively studied, there is missing information on the ecophysiology of early life stages for this species".
L113 (and along the MS): Check the temperature format. There should be an extra space between the number and the unit as you used in L313
L126: Mauffrey et al. 2020? (more than 2 authors)
L142: randomly sampled (change s by d)
L159: Please, clarify why maintenance and measuring temperatures differ between them. I understand given the broad annual variations in temperature a difference of 10 ºC is not too much, but still there is not a justification or a did not find it.
L175: molar instead of Mole?// nitrate NO3- and phosphate PO43- should be written as anions with their respective charges.
L182: Italicize "in situ" and remove the dash "-"
L429: "manage well" instead of "well manage"
L437: replace " is fully performant" by "to perform"
L470: Remove "to" after Beyond
L473: This paragraph should follow the previous sentence.
L548: Replace "subjected to wavy way" by "wave-exposed" (never saw the first expression)
L549: and "also" hydrodynamism.
L550: Replace "Even if" by "Despite"
L553: Replace "stresses" by "stressors"

---

## Round 0.3 · accepted · Accept

I am satisfied with the changes that have been made to the mansucript. This manuscript is ready for acceptance in PeerJ. Thank you for your contribution.